# Nondestructive Detection of Codling Moth Infestation in Apples Using Pixel-Based NIR Hyperspectral Imaging with Machine Learning and Feature Selection

**DOI:** 10.3390/foods11010008

**Published:** 2021-12-21

**Authors:** Nader Ekramirad, Alfadhl Y. Khaled, Lauren E. Doyle, Julia R. Loeb, Kevin D. Donohue, Raul T. Villanueva, Akinbode A. Adedeji

**Affiliations:** 1Department of Biosystems and Agricultural Engineering, University of Kentucky, Lexington, KY 40546, USA; nader.ekramirad@uky.edu (N.E.); alfadhl.alkhaled@uky.edu (A.Y.K.); ledo235@g.uky.edu (L.E.D.); jrlo244@g.uky.edu (J.R.L.); 2Department of Electrical and Computer Engineering, University of Kentucky, Lexington, KY 40546, USA; kevin.donohue1@uky.edu; 3Department of Entomology, University of Kentucky, Princeton, KY 42445, USA; raul.villanueva@uky.edu

**Keywords:** apples, codling moth, hyperspectral imaging, near-infrared, machine learning, feature selection

## Abstract

Codling moth (CM) (*Cydia pomonella* L.), a devastating pest, creates a serious issue for apple production and marketing in apple-producing countries. Therefore, effective nondestructive early detection of external and internal defects in CM-infested apples could remarkably prevent postharvest losses and improve the quality of the final product. In this study, near-infrared (NIR) hyperspectral reflectance imaging in the wavelength range of 900–1700 nm was applied to detect CM infestation at the pixel level for three organic apple cultivars, namely Gala, Fuji and Granny Smith. An effective region of interest (ROI) acquisition procedure along with different machine learning and data processing methods were used to build robust and high accuracy classification models. Optimal wavelength selection was implemented using sequential stepwise selection methods to build multispectral imaging models for fast and effective classification purposes. The results showed that the infested and healthy samples were classified at pixel level with up to 97.4% total accuracy for validation dataset using a gradient tree boosting (GTB) ensemble classifier, among others. The feature selection algorithm obtained a maximum accuracy of 91.6% with only 22 selected wavelengths. These findings indicate the high potential of NIR hyperspectral imaging (HSI) in detecting and classifying latent CM infestation in apples of different cultivars.

## 1. Introduction

Apples are a very important fruit in the global produce market and industry. The United States of America is the second largest producer of apples, producing about 4.5 million tons of apples in 2020 [1], exporting 1 out of 3 apples grown, and averaging $1 billion annually on apple exports [2]. Additionally, apples are the most consumed fruit in the US, with the market value of about $5 billion in 2018 [2]. Because apple marketing is such a big business worldwide, preserving their quality to meet the ever increasing demands of consumers is essential. Codling moth (CM) *Cydia pomonella* (Lepidoptera: Tortricidae) is known to be the most devastating pest that infects apples [3,4]. It causes direct damage to the fruit’s skin and pulp. CM is known to infest pome fruits, with a special preference for apples in almost every country the fruit is grown [3]. This larva enters the apple by feeding through the skin of the fruit, burrowing into the fruit’s core to cause major damage [4]. If untreated, CM can result in up to a 50% loss in pre- and post-harvest apples [3]. Furthermore, production will only tolerate 1% of affected fruit [4], where if any apple infestation is found in some of the US’ top importing countries, the whole shipment is rejected [5]. Detection of infestation, therefore, is very critical but the current manual random methods are inefficient.

Presently, apple quality assessment, including testing for possible insect infestation, is done at random, manually, and in a destructive manner. When assessing apples for packaging, inspectors visually examine the external qualities, scoring the apple surface to comply with certain specifications and tolerances for defects. To determine the internal quality, apples are cut in half to visually inspect their cross-sectional areas [6]. After testing, the used apples are discarded, wasting about three percent of the product [7]. In this way, detection of infestation is time-consuming, costly, subjective, and laborious, and yet does not assure that the batch will be pest free. Currently, machine vision has been implemented to monitor the outside of the apple at low-cost and rapid speed, but issues arise in interference of the sample’s color and the presence of stem and calyx [8], coupled with the inability to inspect the internal qualities of the apple where most pest infestation damage resides. To correct this issue and increase detection efficiency, a better method is needed to identify both internal and external damages to apples by a pest such as CM.

Whereas current techniques are very wasteful, nondestructive techniques preserve the fruit, give a definitive result, and can easily look at the whole batch to ensure no bad apple gets through to the supply chain. Using forms of nondestructive testing to assess certain qualities of apples is not new. Some of the nondestructive techniques used on apples include hyperspectral imaging (HSI) [9,10], vibro-acoustic signaling [11,12], ultrasonic acoustic detection [13], delta absorbance meter [14], machine vision [15,16], and spectroscopy [10,17]. Each technique has its advantages and disadvantages; however, the one that stands out is HSI.

While some of these techniques have been used to detect and classify infested apples, such as the work done with acoustic emissions [18] and vibro-acoustic signaling [12], HSI is ideal because it conveys additional useful information for nondestructive applications. HSI combines the capability of spectroscopy and machine vision techniques. Spectroscopy is used to create a spectrum of data based on light absorbance at different wavelengths [19,20]. This is useful in finding specific chemical components but lacks a sense of location or direction since the device scans at a single point [20,21,22]. Additionally, machine vision converts photographic scanning of 3D objects into 2D images by capturing and documenting the reflected light into grayscale and RGB color [23]. Machine vision is great at scanning objects quickly and acquiring a sense of location, allowing for analysis of spatial qualities such as size, shape, and color [24,25]. However, it only looks at the surface of the object in primary color [20]. HSI uses the best parts of both techniques, looking at the reflectance at every point of the image showing a spectrum of reflectance for each pixel in the spatial image while still retaining the analytical benefits of the two techniques [26]. Each hyperspectral image is a three-dimensional data cube (3D hypercube) with X and Y coordinates as the spatial information and *λ* as the spectral data. HSI not only has the capability to detect infestation on and in the sample under test, but also is used to find the exact location of infestation due to the spatial information and the ability to evaluate the different levels of pixels in the images [27,28].

HSI has been investigated as a rapid and relatively low-cost nondestructive technique in the quality assessment of apples. This application mainly falls into three categories including external quality, internal quality, and pest detection [29,30]. Regarding external quality of apples, HSI was used to evaluate defects (e.g., surface defects and bruising) because of its ability to penetrate beneath the apple’s skin. For example, bruises on apples are detectable in the range of shortwave-near-infrared (SNIR), particularly around 675 nm and 960 nm, which represent the region for carotenoids, chlorophyll pigments, sugar, and water content [31]. This reveals that the bruise on the apple causes large imbibition of water and total sugar contents at the early stage of bruising, and then causes assimilation of chlorophyll pigments and carotenoids in the subsequent stage [32]. Thus, the application of HSI to detect bruises on apples can reduce or prevent further losses from cross-contamination of others by damaged apples. HSI also has been used for nondestructive prediction of internal quality of apples such as the nutritional value, texture, and flavor components, and in estimating physiochemical parameters such as vitamin and sugar content [33]. Additionally, HSI has been tested for safety assurance of apples through effective detection of pests [30,34]. In 2021, Ekramirad et al. [30] determined the best classification result for CM detection in apples using NIR HSI and the mean spectra extraction method on dataset consisting of three apple orientations (clayx, sides and bottom). Applying partial least squares-discriminant analysis (PLS-DA) classifier, they obtained an overall validation accuracy of 81.04%. While the calyx and side orientations had similar classification rates of about 80%, the stem orientation gave the lowest classification accuracies. These results are better than the findings of Rady et al. [34] who achieved a maximum classification rate of 74% using the side orientation of apples, using all the spectral wavelengths in the Vis-NIR range. However, they reported that by reducing the data dimensions using the sequential forward selection method, their classification accuracy was enhanced to 82%. In the same study conducted by Ekramirad et al. [30], they applied a second pixel-wise method instead of the mean spectra extraction and found an accuracy as high as 98.2% in classifying the infested and healthy pixels using the random forest (RF) classifier. However, they had manually segmented a rectangular ROI around the calyx end of the apple sample to extract pixels data by spectra for infested and healthy, which is a cumbersome subjective task that increases the processing time, hence there is a need to develop a new method for automatic extraction of target pixels. Additionally challenges to be considered in developing an automatic algorithm based on HSI for the classification of CM-infested apples include the following. First, since the shape and size of ROI in HSI affect the measurement performance [35], a proper geometry should be selected for the ROI. Second, the infested region should be well localized for accurate labeling as an infested class. Therefore, to address these issues a special procedure was developed in this study to automatically extract pixel-wise ROI around the calyx of the apples, which is the usual point of entry for the CM larvae.

Normally, HSI technique produces large spectral data and for this, its analysis is associated with the utilization of mathematical or statistical methods to make it readable and to discover useful information about the data. However, such analyses are time consuming because of the large size of the datasets. In computational intelligence methods, dimension reduction is often used to optimize data processing time, reduce dimensionality, and enhance data generalization [36,37]. Principal component analysis (PCA) is the main dimensional reduction step used for hyperspectral data to transform its spectra into some independent features. Moreover, some approaches have been developed to optimize the HSI to perform real-time hyperspectral data reduction using the extraction of predefined features [38,39]. This is carried out by real-time multiplication of the acquired spectral data by a feature extraction operator (vector-to-scaler) consisting of the desired features (less than ten as opposed to >100 spectral features in the hypercube) predefined by experiment. The optimal feature extraction operators are usually obtained by mathematical and statistical methods, such as that applied in PCA. Some of the most frequently used classification methods include k-nearest neighbor (kNN), Linear discriminant analysis (LDA), Quadratic discriminant analysis (QDA), and Naïve Bayes (NB) [40]. In addition, PLS-DA has been proven in many studies to be a powerful classification method for HSI data analysis with high-dimensional data [41]. Additionally, the ensemble methods such as RF and gradient tree boosting (GTB) can integrate weak classifiers to achieve powerful anti-noise classifiers [42].

While HSI has been widely applied for quality assessment of agricultural products, there is no report on the application of NIR hyperspectral imaging combined with feature selection algorithm and various machine learning algorithms to detect CM latent infestation in apples. The infestation of plant tissue by pests can induce different defense mechanisms, such as hypersensitive reactions, production of metabolites and proteins, and altered plant tissue structure, leading to various reflectance spectral signatures that can be measured and localized by spectral imaging methods [43]. Thus, the main objective of the current work was to develop and validate a robust model for the accurate detection of latent CM infestation in apples based on the NIR HSI technique. The specific objectives were to: (1) develop an automatic procedure for pixel-based extraction of infestation region on apple to address the issues related to manual segmentation of the infested area, (2) compare the results of the classification method for three major apple cultivars of Fuji, Gala, and Granny Smith, and (3) select some optimal wavebands for reducing the dimensions of the large scale HSI data leading to a multispectral imaging system.

## 2. Materials and Methods

### 2.1. Sample Preparation

The apple samples used in the experiment were USDA-certified organic Gala, Fuji, and Granny Smith cultivars purchased from a commercial market in Princeton, KY, USA in October 2020. After careful inspection, 60 sample apples similar in size, diameter, and shape, and without infestation and mechanical damage, were chosen from each cultivar. The apples were then disinfected against fungal and bacterial decay by washing in a 0.5% (*v*/*v*) sodium hypochlorite solution [44]. The samples were rinsed with distilled water and dried in open air at ambient conditions in the laboratory (Department of Entomology, University of Kentucky, Princeton, KY, USA). To artificially infest the apples, newly hatched neonate of CM larva was placed near the calyx end of each apple in an isolated cup (8 cm bottom diameter, 10 cm top diameter, 10 cm high) with a plastic lid. Apples of each cultivar were divided into 20 control and 40 infested groups and stored in an environmental control chamber at 27 °C and 85% relative humidity for three weeks to cause infestation to occur. The hyperspectral data acquisition was carried out in the Food Engineering lab at Biosystems and Agricultural Engineering Department, University of Kentucky, Lexington, KY, USA.

### 2.2. HSI System and Image Acquisition

A HSI system based on shortwave near-infrared (NIR) bands was used to acquire the hyperspectral data of all apple samples—control and infested (Figure 1). The HSI system consisted of a NIR spectrograph with a wavelength range from 900 nm to 1700 nm and a spectral resolution of 3 nm (N17E, Specim, Oulu, Finland), a moving stage driven by a stepping motor (MRC-999-031, Middleton Spectral Vision, Middleton, WI, USA), a 150 W halogen lamp (A20800, Schott, Southbridge, MA, USA), an InGaAs camera (Goldeye infrared camera: G-032, Allied Vision, Stradtroda, Germany) mounted perpendicular to the sample stage and a computer with data acquisition and analysis software (FastFrame™ Acquisition Software, Middleton Spectral Vision com, Middleton, WI, USA). Three scanning orientations of the stem, calyx, and side of each apple were captured during hyperspectral image acquisition. To acquire clear images, the parameters of the sample stage speed, the exposure time of the camera, the halogen lamp angle, and the vertical distance between the lens and the sample were set to 10 mms^−1^, 40 ms. 45°, and 25 cm, respectively. Samples were placed on the sample stage and captured in a line scanning or pushbroom mode. The acquired hyperspectral images contained 256 wavelength bands as “*.raw” file along with a header file as “*.hdr”.

### 2.3. Preprocessing of Hyperspectral Images

#### 2.3.1. Image Calibration

This is needed to correct the acquired raw images with the white and dark reference images to eliminate the influence of illumination and dark current of the camera. The reference image data were obtained after the samples were scanned every day. The dark reference images were obtained by completely covering the lens of the HSI system while turning off the lights. The white reference images were acquired using a polytetrafluoroethylene (PTFE) Teflon plate of 99% reflectivity and 10 mm thickness placed on the black sample stage. The calibration was done based on the following equation:(1)$$R=\frac{R_{0\;}-R_{d}}{R_{w\;}-R_{d}}$$
where *R*_0_ is the raw hyperspectral image, *R*_d_ is the dark image, and *R*_w_ is the white reflectance image [45,46].

#### 2.3.2. Infestation Region Acquisition

After the acquisition and correction of the hyperspectral images, the spectral information of the infested and healthy tissue was automatically extracted from ROIs using the algorithm described in Figure 2. Since the CM larvae, especially the first generation, mostly enter apples from the calyx end [47] and the initial results by Ekramirad et al. [30] showed that the highest infestation classification accuracy achieved in images from the calyx view, the ROI to extract infested pixels was segmented around the calyx end. This novel method can select the complete infested region with pixels in the healthy region as few as possible to obtain a precise infested region for subsequent classification. To do this, first the background and calyx end were segmented out using the image at 1084 nm wavelength to obtain a masked image in a binary image format using a binary thresholding method. To obtain a solid area around the calyx, the morphological image processing of erosion operation was applied. Then, the center of the calyx area was localized using mathematical operations to calculate the centroid of the eroded area. Finally, having the orientations of the center of the calyx area, a circular region with 50 pixels diameter was drawn as a mask binary region. The circular ROI fits with the spherical shape of apple fruit and it has been shown that a round ROI gave higher accuracy and predictive capability than square ROI in HSI on apples [35]. Then, the circular masked image was applied on each image of the hypercube (i.e., all the 256 wavebands) to obtain the calyx area with other pixels equal to zero. The spectrum for each pixel in the circular ROI was then extracted and unfolded and labeled for building the dataset to develop machine learning models.

#### 2.3.3. Spectral Extraction and Preprocessing

To obtain the spectral characteristics of apples, the spectral for each pixel inside the ROI was extracted in the form of reflectance intensity versus wavelength and then labeled as either infested or healthy spectral signature. After spectral data extraction, pre-processing was carried out by wavelength trimming, maximum normalization, a Savitzky-Golay smoothing filter, and mean centering to remove the noisy wavelengths at the edges of each spectrum, to get all data to the same scale, to account for particle size scattering and path length difference effects, and to keep only significant features, respectively. The maximum normalization was carried out by dividing each spectrum by the maximum value [29]. The Savitzky-Golay method involves the application of the second-order polynomial and the filter window of length 31.

#### 2.3.4. Dimensionality Reduction

With respect to data architecture in HSI, high dimensional images with fixed training sample size can result in overfitting problems leading to degraded classification rates. It is usually the case when the size of training samples is limited in comparison to the feature space size resulting in low generalization of the results and overfitting problems [48]. The higher the dimensionality of the model, the higher the likelihood of over-fitting. Therefore, hyperspectral data size compression, especially spectral dimensionality reduction is usually required to achieve better data visualization, save storage space, eliminate redundant data, and avoid model over-fitting. Principal component analysis (PCA) was used in this study as the dimensionality reduction technique. PCA is a transformation of the data through an axis rotation, in the direction of maximum variance. Successive (principal components) PCs are the linear combinations of the variables with maximum variance, which are orthogonal to the previously computed components. The total variance can be represented in a significantly small number of components (extracted features). The PCs are the eigenvectors of the covariance matrix of the data, and the associated variance is represented in the corresponding eigenvalues. The PCs are orthogonal and have successively ordered variances. PCA transforms multivariate data into a new coordinate system to produce new uncorrelated orthogonal variables which are called PCs or loadings. These PCs are arranged according to their eigenvalues, with the 1st PC having the highest variance, the 2nd PC containing the highest residual variance, and so on [49]. As most of the information is included in the first PCs, eliminating PCs with a small variance will remove unnecessary information. The advantages of this technique over nonlinear dimensionality reduction techniques include being easy to apply, invertible, and volume-preserving transformation.

#### 2.3.5. Spectral Variable Selection

Acquired data from HSI systems usually have high dimensions both spatially and in spectral form. These data contain highly correlated continuous wavelengths with a lot of redundancy resulting in high complexity and computation costs. Feature selection methods through selecting optimal wavelengths can provide the informative wavelengths to build fast and simpler multispectral imaging models. In this study, the optimal wavelength selection was conducted using the sequential stepwise selection method. In this method, which is a wrapper method, a specific machine learning classifier is fitted to the dataset. It acts as a greedy search approach through evaluating all the possible combinations of features against the evaluation criterion, which is the performance measure such as accuracy, precision, sensitivity, etc. Finally, it selects the combination of optimum features that gives the best results for the specified machine learning algorithm.

### 2.4. Development of Machine Learning Classifiers

Having the spectral data for healthy and infested pixels from all the samples, the labeled dataset was built by organizing pixels (observations) as rows, and the features (spectral data) as columns. The predictor variables (features) are the spectral data while the dependent variables are the classes, namely control and infested. After building the dataset, Kennard & Stones algorithm was implemented to split 70% of the data as the training and 30% as the validation datasets. The Kennard & Stones method has been widely used in chemometrics and spectroscopy, and it has been proven to give good performance in separating spectral data into training and test sets [50]. Different machine learning classification algorithms including Linear discriminant analysis (LDA), k-Nearest Neighbors (kNN), PLS-DA, and two ensemble methods, namely Random forest (RF) and GTB, were performed and compared for their classification accuracies. For evaluating the performance of the various models in this study, five-fold cross-validation was used. The metrics used for assessing the classification performance of the models included precision, recall, and the F1-score. Precision is the positive predicted value and quantifies the correctly classified pixels as infested (the fraction of true positives out of all positive predictions whether true positive or false positive). While precision gives a quantitative measure of how exact the classifier’s prediction is, recall (or sensitivity) helps avoid missing any undetected infested samples. Recall is the true positive rate that relates to the number of pixels belonging to the infested area that were predicted as positive (true positive) and those that the model incorrectly does not capture as infested (false negative). F1-score is the harmonic mean of recall (R) and precision (P), calculated as: F1 = 2RP/(R + P), reflecting the balance between the classifier’s precision and recall [51]. The F1-score can be used to evaluate the entire model, considering both the precision and recall, making it a sensitive metric to changes in the data distribution and ratios.

All algorithms used in this study for pre-processing and data analysis and post-processing were performed on Python 3.7 (Python Software Foundation, https://www.python.org/ (accessed on 15 October 2021)) platform and in Jupyter Editor Notebook. The open-source libraries of spectral, Numpy, Scikit-learning, and Matplotlib were used in this work.

## 3. Results and Discussion

### 3.1. Spectral Analysis

Figure 3 shows the typical mean spectra for healthy and infested apple samples as normalized reflectance versus wavelength. While the average spectra of the control and the infested samples have a similar trend and curve variation tendency, the reflectance of the healthy samples is remarkably higher than the one for the infested samples (or the absorbance of the infested samples are higher than the healthy ones). Thus, this NIR reflectance difference for healthy and infested samples shows the potential to be applied for the binary classification. As reported by several authors [26,52,53], the absorbance of defective samples was higher than the healthy ones due to the cellular structure difference. As a result of plant tissue infestation, there will be biochemical, tissue structure, and pigment composition changes, leading to the different spectral signatures [54].

As shown in Figure 3, there are some distinct absorption valleys around 950, 1200, and 1400 nm in the mean spectra of both sample classes. The absorption at about 950 and 1200 nm relates to the first overtones of O-H band in water molecules [55,56]. The absorption at around 1400 nm is attributed to the combination of the second overtone of C-H and the first overtone of O-H [9]. The spectral curve of samples in this study agrees with the finding of other studies on apples in the same spectral range [9].

### 3.2. Pre-Processing and Feature Extraction Results

For classification of infested samples of the three different cultivars of apples, PCA was performed on the preprocessed spectra before building the classification models. Results showed that the sum of the variance explained by the first three PCs for all the cultivars was more than 98% of the total variance. It means the sum of the accumulative contribution rate of the first three PCs represents 98% of the total variability of the spectral data, so it could be a reasonable way to recognize patterns in the tested samples using these limited number of PCs. As shown in Figure 4, the control and infested samples were well clustered with some minor overlaps between them. Moreover, PCA score values for infested apples were tightly clustered over the two first PCs space, while scores for control fruits were more widely scattered. Therefore, the machine learning models for classification of apples were built using the extracted PCs as features. Moscetti et al. [57] reported similar PCA score plot trends for non-infested and infested olive fruits using the NIR spectroscopy for the mean spectra of the whole fruit where the first two PCs accounted for 98.3% of the total variance. They used the pre-processing steps of multiplicative scatter correction, a Savitzky-Golay smoothing filter, and mean centering followed by PCA dimensionally reduction and LDA, QDA, and kNN classification. Additionally, the results of Keresztes et al. [29] for pixel-based apple bruise detection using shortwave infrared HSI showed that the three first PCs represented 98.36%, 1.24%, and 0.15% of the total variance in the data. They also used the pre-processing methods of multiplicative scatter correction, Savitzky-Golay smoothing, and mean centering before PCA dimensionally reduction followed by PLS-DA classification.

### 3.3. The Results of Machine Learning Classification

In order to compare the results of different approaches for classification of apple samples, classification results of three approaches are presented in this section. Table 1 provides the classification results of infested and healthy Fuji apples using the mean spectra extraction method and for three orientations of stem, calyx, and side of apple along with the data for all the orientations together. In this method the reflectance spectrum for each pixel was extracted and then the average of all the pixels were calculated as the mean reflectance spectra for that sample. As shown in Table 1, the best classification result for the mean spectra extraction method was achieved using the data for all the orientations of apples and by PLS-DA and RF classifiers, with a validation accuracy of 92%. While the calyx and side orientations had similar classification rates of 88.9%, the stem orientation gave the lowest classification accuracies. These results are better than the findings of Rady et al. [34] who achieved a maximum classification rate of 74% using the mean reflectance spectra from Vis-NIR hyperspectral images of the side views of apples.

In the second approach, pixels from infested apples were localized and segmented manually using a 10 × 10 rectangular ROI around calyx end of samples. To do this, the ROI were selected in the images and the spectrum of each pixel in the ROI was extracted. Thus, a total of 100 spectra for each infested or control apple were extracted and labeled to build the classification dataset. Table 2 shows the classification results for the control and infested pixels in apples using the manual ROI selection method. The result gave a good performance with the accuracy of up to 99.24% for the ensemble classifiers. These findings are in good agreement with those of Munera et al. [53] who reported an overall accuracy of up to 97.5% in classifying the healthy and defective pixels in hyperspectral images of loquat fruits using the manual ROI selection method.

### 3.4. Performance of Classification Models Based on Apple Cultivar

The detailed performance of different models for classification of CM-infested and healthy samples of three apple cultivars, namely Gala, Fuji, and Granny Smith using the automatic pixel-based method developed and implemented in this study is shown in Table 3. All the classifiers gave higher results using the PCs as the input variables except for LDA and PLS-DA, which gave slightly better classification rates using raw data without dimensionality reduction. The GTB ensemble method yielded the highest classification rates for all three cultivars, reaching as high as 97.4% accuracy of the validation set for Fuji apples. Similar classification rates were achieved by Saranwong et al. [58] using HSI in the range of 400–1000 nm in reflectance mode to assess fruit fly larvae infestation in mango. Using a discriminant analysis classifier, they obtained a validation classification rate of up to 99.1% and 94.3% for infested and healthy fruits, respectively. Haff et al. [59] also researched the same insect in mango using the same method and the classification rates reached 99% for infested samples. While they achieved a high classification rate in both studies, they artificially created pores on the fruit in a grid pattern to expose the fruit to the pest insects to have a priori knowledge of the locations of infestation. Then they extracted the spectra from the pore locations and compared them with the spectra from healthy areas to identify the spots generated in hyperspectral images of mangoes infested with fruit fly larvae. They reported that classifying the samples which were deliberately infested following a predefined pattern, and the algorithm relying on that pattern in the images would be useless in real-world applications. In a similar study to our current research, Rady et al. [34] studied the ability of Vis-NIR HSI (400–900 nm) in the reflectance mode for the detection and classification of CM infestation in GoldRush apples. Their best classification rates were obtained using decision trees at five selected wavelengths with an overall classification rate of 82%. Their relatively low classification rate can be related to the limited spectral range used to detect internal and invisible defects in samples. Moreover, they used the traditional image processing-based method combined with the mean spectral extraction for the whole sample. Thus, the broader spectral range as well as the pixel-based method for extracting the spectral signature of infested regions could be the reason behind the higher classification rates in the current study. In another study on apples for a different application, Che et al. [42] used pixel-based Vis-NIR HSI to classify the bruised Fuji fruits. They also reached their best accuracy of 99.90% with the ensemble method.

Table 4 summarizes some important performance evaluation indices for the best classifier (GTB) for the three apple cultivars. The most important metric for the detection of a pest of concern such as CM is recall or sensitivity which reflects the amount of incorrectly classified infested samples (false negative) or the truly infested samples that were not detected as infested and were classified as healthy. As it is shown in Table 4, the recall values for infested samples are higher than the precision values for all the cultivars reaching as high as 0.98 (98%).

### 3.5. Optimal Wavelength Selection

As mentioned above, the optimal wavelengths were selected from the whole spectra by the sequential forward selection (SFS) method to minimize variable collinearity and select the most informative variables. This algorithm started with one wavelength and then added a new one in each iteration process, and a specified number of wavelengths were selected at the end. The selections of optimal wavebands are shown in Figure 5 and Table 5. The results in Figure 5, obtained by applying SFS, illustrate a graph of classification accuracy changing with increasing the number of selected wavelengths. As shown, when 22 wavelengths variables were selected, the classification performance rate approaches an asymptote while the number of selected variables was significantly less than the raw spectral data (356 wavebands). Therefore, the optimal variable wavelength subset, which consists of 977.2, 983.9,1050.9, 1064.3, 1081.0, 1151.28, 1184.6, 1228.0, 1248.1, 1288.1, 1351.4, 1447.9, 1530.9, 1554.2, 1574.1, 1590.7, 1627.1, 1647.0, 1653.7, 1657.0,1663.6, and 1680.2 nm, was determined for classifying the CM infestation on Fuji apples, while the corresponding number of sampled variables was 22. The first 22 wavelengths for apples are mainly distributed around 1000, 1200, 1550, and 1650 nm.

After selecting optimal wavelengths by SFS, the selected optimal wavelengths carrying the most valuable information in the spectra were considered as the input variables to build the ensemble classifier model. Additionally, to further evaluate the representativeness of the chosen optimal size of the validation set, the classification results of the ensemble model based on the different number of optimal wavelengths for each of the three cultivars were compared (Table 5).

The classification accuracy of 91.6% for the validation set, which is very close to the maximum classification accuracy obtained with the whole range of wavelengths (97.4%), further validated the representativeness of the chosen optimal size of the data set. While this result is better than the results of Rady et al. [34] who achieved an accuracy of 82% in classification of CM-infested apples using Vis-NIR HSI, their best classification accuracy was obtained using only five selected wavelengths. It is worth noting that the number of wavelengths in current study reduced from 356 to 22, which only accounts for 6.17% of the total wavelengths, making the simplified model better than the model developed using the full spectra. Overall, results indicate that this is an effective way to select optimal wavelengths to build discriminant models by SFS, with a potential reduction in computational cost and relatively satisfying model performance.

## 4. Conclusions

In this study, machine learning models were developed to perform classification of CM infestation in apples using pixel-level NIR hyperspectral image data. Combined NIR HSI, machine learning and image processing methods were applied to discriminate healthy and the infested tissues in three apple cultivars. The results of three approaches were provided; the first approach was based on using image-level mean spectra extraction for the whole sample analysis, and the second and third approaches were conducted at the pixel level using manual and automatic ROI segmentation around the infested area of the sample, respectively. Furthermore, the optimal wavelengths were selected using SFS algorithm to develop multispectral models. The total classification accuracy for the infested and healthy samples are as high as 97.4% for the validation dataset using GTB ensemble classifier among others. The feature selection algorithm obtained a maximum validation accuracy of 91.6% with only 22 selected wavelengths. Therefore, the NIR HSI method for infestation detection demonstrated the capacity to detect CM infestation in apples of different varieties with potential in post-harvest inline apple sorting applications. Overall, the good results obtained in this study represent a promising step forward for sorting technologies employed in the apple processing units especially in packinghouses and export/import inspections. Moreover, the proposed NIR HSI could be useful as a remote monitoring tool for quality control and for studying CM incidence directly in the orchard; for example, through UAV-based HSI. Finally, future research could include evaluating the computational costs and processing speed, improving hardware, and applying other machine learning methods such as deep learning, as these could improve the accuracy and the robustness of the HSI detection system.

## Figures and Tables

**Figure 1 foods-11-00008-f001:**
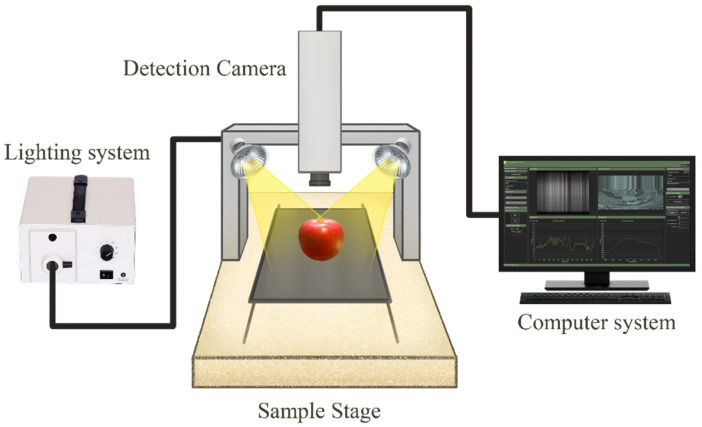
Schematic of the hyperspectral imaging system.

**Figure 2 foods-11-00008-f002:**
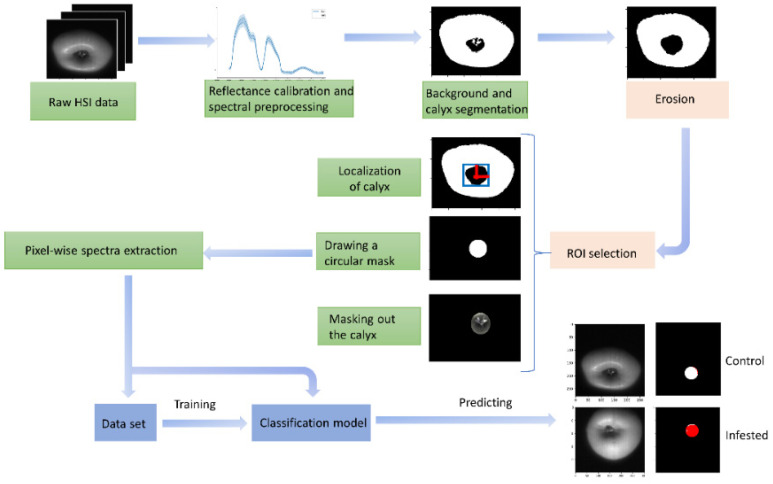
Flowchart of apple infestation area acquisition around calyx end for building the classification model. HIS: hyperspectral imaging; ROI: region of interest.

**Figure 3 foods-11-00008-f003:**
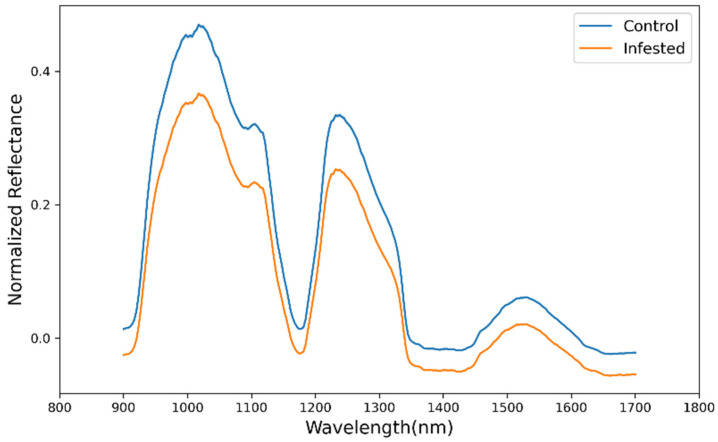
Mean reflectance spectra of control and CM-infested samples acquired by near-infrared hyperspectral imaging (NIR HIS). CM: codling moth.

**Figure 4 foods-11-00008-f004:**
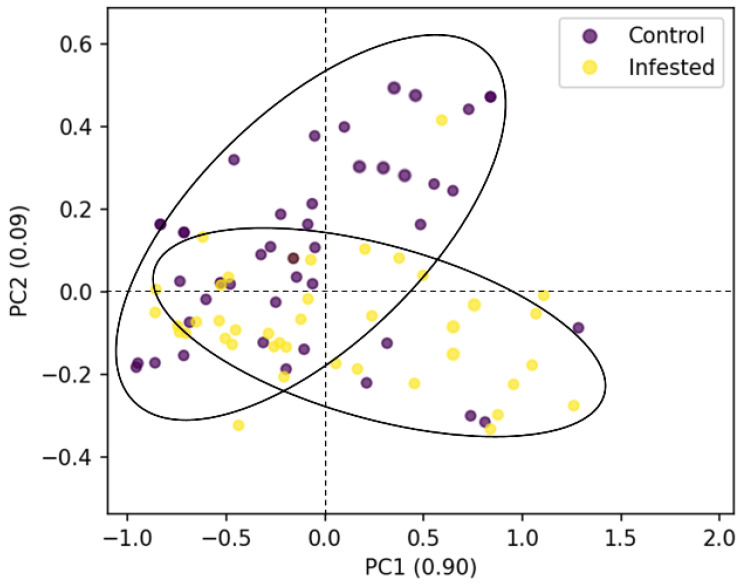
Principal component analysis of two types of apple sample tissues for Fuji cultivar computed from the mean spectral of the whole fruit.

**Figure 5 foods-11-00008-f005:**
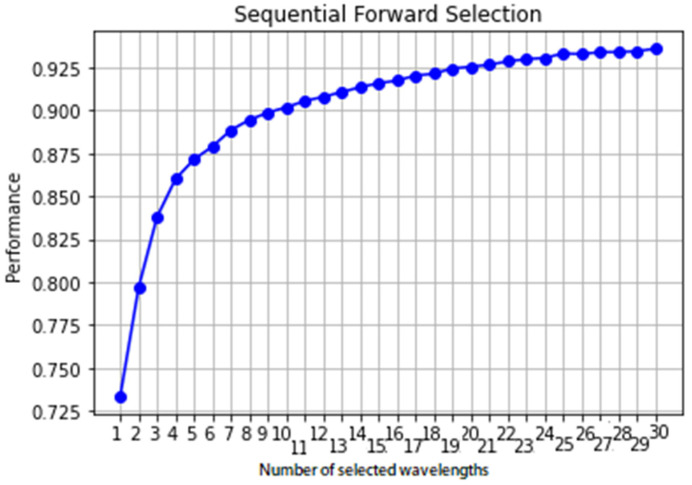
Classification performance (accuracy) as a function of the number of wavelengths.

**Table 1 foods-11-00008-t001:** Results of the PCA-based classification of control and infested samples for training and validation sets based on mean spectra for each sample.

Sample Orientation	Classifier ^1^	Training Set (%)			Validation Set (%)		
		Precision	Recall	Total Accuracy	Precision	Recall	Total Accuracy
Stem	LDA	95.00	94.00	94.70	57.00	58.00	62.50
	kNN	58.00	57.00	57.90	36.00	42.00	62.50
	**RF**	**100**	**100**	**100**	**83.00**	**92.00**	**87.50**
	AdaBoost	95.00	95.00	94.70	75.00	83.00	75.00
	**PLS-DA**	**100**	**100**	**100**	**83.00**	**92.00**	**87.50**
Calyx	LDA	90.00	90.00	90.50	78.00	78.00	77.80
	kNN	63.00	61.00	61.90	68.00	68.00	67.00
	RF	100	100	100	88.00	83.00	83.30
	**AdaBoost**	**100**	**100**	**100**	**92.00**	**88.00**	**88.90**
	PLS-DA	90.00	90.00	90.50	78.00	78.00	78.00
Side	LDA	100	100	100	83.00	80.00	77.80
	kNN	86.00	80.00	80.00	75.00	60.00	55.60
	RF	100	100	100	83.00	80.00	77.80
	AdaBoost	100	100	100	83.00	80.00	77.80
	**PLS-DA**	**100**	**100**	**100**	**90.00**	**90.00**	**88.90**
All	LDA	80.00	80.00	79.00	71.00	73.00	72.00
	kNN	76.00	76.00	76.30	70.00	71.00	72.00
	RF	100	100	100	91.00	94.00	92.00
	AdaBoost	100	100	100	88.00	91.00	88.00
	**PLS-DA**	**98.00**	**98.00**	**98.00**	**91.00**	**94.00**	**92.00**

^1^ LDA: Linear Discriminant Analysis, kNN: k-Nearest Neighbors, RF: Random Forest, PLS-DA: Partial Least Squares-Discriminant Analysis. Bolded line indicates the best result.

**Table 2 foods-11-00008-t002:** Results of the PCA-based classification of control and infested samples for training and validation data sets based on manually selected ROI.

Classifier ^1^	Training Set (%)			Validation Set (%)		
	Precision	Recall	Total Accuracy	Precision	Recall	Total Accuracy
LDA	72.20	79.20	75.24	71.60	78.40	74.64
kNN	100	99.20	99.52	99.60	98.80	99.06
RF	100	100	100	99.20	99.60	99.24
AdaBoost	100	100	100	98.00	98.4	98.20
PLS-DA	84.60	88.80	86.40	80.60	82.60	80.18

^1^ LDA: Linear Discriminant Analysis, kNN: k-Nearest Neighbors, RF: Random Forest, PLS-DA: Partial Least Squares-Discriminant Analysis.

**Table 3 foods-11-00008-t003:** Classification accuracy (%) for validation data set based on automatically selected pixels for three apple cultivars.

Classifier	Raw Data (No Dimensionality Reduction)				PCA-Based			
	Gala	Granny Smith	Fuji	All	Gala	Granny Smith	Fuji	All
LDA	65.38 ± 0.62	72.24 ± 0.23	70.46 ± 0.72	69.22 ± 0.10	65.38 ± 0.62	70.38 ± 0.17	66.94 ± 0.33	68.70 ± 0.14
SVM	80.18 ± 0.06	76.42 ± 0.17	81.40 ± 0.44	72.54 ± 0.36	82.60 ± 0.70	77.20 ± 0.18	81.62 ± 0.33	73.84 ± 0.39
kNN	93.72 ± 0.19	93.26 ± 0.15	95.46 ± 0.32	89.12 ± 0.12	93.80 ± 0.15	93.30 ± 0.07	95.69 ± 0.26	88.84 ± 0.11
RF	89.66 ± 0.19	89.04 ± 0.18	91.52 ± 0.27	82.82 ± 0.14	94.28 ± 0.31	93.22 ± 0.25	96.62 ± 0.13	89.74 ± 0.13
GTB	92.32 ± 0.37	91.00 ± 0.25	94.68 ± 0.39	84.66 ± 0.18	94.76 ± 0.16	93.66 ± 0.18	97.36 ± 0.28	90.00 ± 0.23
PLS-DA	62.76 ± 0.66	71.64 ± 0.24	68.56 ± 0.15	69.14 ± 0.15	62.76 ± 0.66	71.34 ± 0.16	66.92 ± 0.35	68.72 ± 0.16

PCA: principal component analysis, LDA: Linear Discriminant Analysis, SVM: support vector machine, kNN: k-Nearest Neighbors, RF: Random Forest, GTB: Gradient tree boosting, PLS-DA: Partial Least Squares-Discriminant Analysis.

**Table 4 foods-11-00008-t004:** Classification performance of gradient tree boosting for control and infested samples for three apple cultivars based on automatically selected pixels for three apple cultivars.

Cultivars	Classes	Precision	Recall	F1-Score	Overall Accuracy (%)
Fuji	Control	0.98	0.96	0.97	97.36
	Infested	0.97	0.98	0.97	
Gala	Control	0.93	0.93	0.93	94.76
	Infested	0.95	0.96	0.95	
Granny Smith	Control	0.91	0.90	0.91	93.46
	Infested	0.95	0.95	0.95	

**Table 5 foods-11-00008-t005:** Classification performance of selected optimal wavelengths.

No. of Wavelengths	Gala		Granny Smith		Fuji	
	Selected Wavelengths (nm)	Classification Accuracy	Selected Wavelengths (nm)	Classification Accuracy	Selected Wavelengths (nm)	Classification Accuracy
30	900.1, 903.5, 920.3, 970.6, 997.4, 100.7, 1014.1, 1071.0, 1077.7, 1261.4, 1278.1, 1281.4, 1298.1, 1324.7, 1328.1, 1361.4, 1384.7, 1408.0, 1447.9, 1464.5, 1447.8, 1477.8, 1627.1, 1647.0, 1653.7, 1657.0, 1663.6, 1666.9, 1676.8, 1693.4	88.5%	900.1, 916.9, 977.2, 1010.7, 1020.8, 1030.8, 1047.6, 1074.3, 1178.0, 1181.3, 1204.7, 1274.7, 1284.7, 1294.7, 1298.1, 1304.7, 1308.1, 1371.4, 1414.6, 1471.1, 1481.1, 1494.4, 1653.7, 1660.3, 1666.9, 1673.5, 1680.2, 1683.5, 1686.8, 1693.4	87.7%	977.2, 980.6, 1044.2, 1074.3, 1077.7, 1081.0, 1137.9, 1147.9, 1151.2, 1211.3, 1264.7, 1294.7, 1314.7, 1344.7, 1348.0, 1381.3, 1421.3, 1507.7, 1530.9, 1544.2, 1560.8, 1580.7, 1623.8, 1630.5, 1647.0, 1650.3, 1653.7, 1657.0, 1663.6, 1673.5	92.4%
22	923.6, 973.9, 1000.7, 1067.6, 1081.0, 1084.4, 1127.8, 1268.1, 1281.4, 1308.1, 1351.4, 1401.3, 1411.3, 1461.2, 1491.1, 1607.3, 1643.7, 1663.6, 1670.2, 1676.8, 1690.1, 1693.4	87.8%	903.5, 916.9, 987.3, 1047.6, 1081.0, 1131.2, 1141.2,1181.3, 1204.7, 1274.7, 1288.1, 1304.7, 1371.4, 1467.8, 1471.1, 1481.1, 1643.7, 1673.5, 1680.2, 1683.5, 1686.8, 1693.4	87.5%	977.2, 983.9, 1050.9, 1064.3, 1081.0, 1151.28, 1184.6, 1228.0, 1248.1, 1288.1, 1351.4, 1447.9, 1530.9, 1554.2, 1574.1, 1590.7, 1627.1, 1647.0, 1653.7, 1657.0, 1663.6, 1680.2	91.6%
15	903.5, 990.6, 997.3, 1071.0, 1084.4, 1281.4, 1294.7, 1371.4, 1384.7, 1447.9, 1477.8, 1663.6, 1673.5, 1680.2, 1690.1	86.2%	1010.7, 1081.0, 1131.2, 1181.3, 1184.6, 1281.4, 1298.1, 1491.1, 1657.0, 1663.6, 1670.2, 1680.2, 1683.5, 1686.8, 1693.4	86.3%	977.2, 983.9, 1050.9, 1074.3, 1081.0, 1311.4, 1381.3, 1401.3, 1447.9, 1507.7, 1627.1, 1637.1, 1647.0, 1653.7, 1673.5	91.0%
5	997.3, 1084.4, 1281.4, 1663.6, 1693.4	81.5%	1014.1, 1274.7, 1494.4, 1683.5, 1693.4	80.7%	983.9, 1050.9, 1311.4, 1653.7, 1663.6	86.2%

## Data Availability

All data used in this project belong to the United State government and the administering institution, University of Kentucky (UK), and can be requested through the UK Library.

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
