# Peer review of "Nondestructive Detection of Codling Moth Infestation in Apples Using Pixel-Based NIR Hyperspectral Imaging with Machine Learning and Feature Selection"

_foods, 2021, doi:10.3390/foods11010008_

Round 1
Reviewer 1 Report
In my opinion, the manuscript should be major revised.
The author mentioned that ''After spectral data extraction, pre-processing was carried out by the trimming of wavelengths, normalization, and Savitzky_Golay smoothing filter to remove the noisy wavelengths at the edges of the spectrum, and to account for particle size scattering and path length difference effects, respectively'', and that was clear the first PC explained 98% of the variance for this kind of complex data is close wrang, the mean centering should be included in the last stage of the used processing steps in order to keep only significant features. The first two PCs were distinguished the two groups, as including the meaning centering enhance that, please explain that in your text. Please construct the new PCA-based classification models as highlighted before. Why the validation models were very low in classification rate, could you explain that? How did you split the data? Can you use Kennard & Stones algorithm to split the data, is more accurate for this kind of data.
The classification models should be compared not only on the accuracy rate but sensitivity, specificity, and total classification rate, please add and discuss. You have used different classification methods, which one is recommended for this application and why? add that in the text and explain the applicability. The same for variable selection techniques, which one is recommended? of course, based on your results add the applicability. The discussion is not very concise and should be reworked. I would suggest revising and comparing those results with appropriate papers and linking those results with applicability (benefits).
In addition, in the conclusion section, a critical evaluation of the advantages and limitations of the existing techniques, the future perspectives, and the expected outcomes in the field of HSI and its applicability in agriculture and food products is necessary and strongly recommended.
Author Response
General Comments
The author mentioned that ''After spectral data extraction, pre-processing was carried out by the trimming of wavelengths, normalization, and Savitzky_Golay smoothing filter to remove the noisy wavelengths at the edges of the spectrum, and to account for particle size scattering and path length difference effects, respectively'', and that was clear the first PC explained 98% of the variance for this kind of complex data is close wrang, the mean centering should be included in the last stage of the used processing steps in order to keep only significant features. The first two PCs were distinguished the two groups, as including the meaning centering enhance that, please explain that in your text. Please construct the new PCA-based classification models as highlighted before. Why the validation models were very low in classification rate, could you explain that? How did you split the data? Can you use Kennard & Stones algorithm to split the data, is more accurate for this kind of data.
Response: We added the mean centering as the last step of the data preprocessing. Before mean centering, the first PC accounted for 91% (not 98%) of the variance. As Figure 4 shows after the mean processing, the first PC accounts for 90% (not 98%) of the variance but the first three PCs account for more than 98% of the variance. Also, the new PCA-based classification models were built as shown Table 1. The performance of the validation models mostly increased after including the mean centering as well as Kennard & Stones splitting suggested by the reviewer.
The classification models should be compared not only on the accuracy rate but sensitivity, specificity, and total classification rate, please add and discuss. You have used different classification methods, which one is recommended for this application and why? add that in the text and explain the applicability. The same for variable selection techniques, which one is recommended? of course, based on your results add the applicability. The discussion is not very concise and should be reworked. I would suggest revising and comparing those results with appropriate papers and linking those results with applicability (benefits).
Response: The sensitivity (Recall) and specificity as well as total accuracy were included in the performance result in Tables 1 and 2. For the results in Table 3 the Recall and specificity were summarized in Table 5 for the best method.
Overall, the ensemble methods and PLS-DA gave the highest accuracy. However, the ensemble method was recommended for this application since it combines weaker classifiers to build powerful and anti-noise methods for these high-dimensional spectral data.
Results of some appropriate papers was added in the discussion to compare their results with the current work (lines 350-359 and 427-429)
In addition, in the conclusion section, a critical evaluation of the advantages and limitations of the existing techniques, the future perspectives, and the expected outcomes in the field of HSI and its applicability in agriculture and food products is necessary and strongly recommended.
Response: Included in the conclusion section.

Reviewer 2 Report
The manuscript is well structured, clear and the conclusions are in accordance with the objectives. The analysis of the data is correct as well as the discussion of the results.
There is one issue that is not clear:
Recall has been defined in line 272 as " Recall is the true positive rate that relates to the number of pixels belonging to infested area that were predicted positive and those that the model incorrectly does not capture as infested. ".
But in line 381, it states: "The most important metric for the detection of a pest of concern such as CM is recall or sensitivity which reflects the amount of incorrectly classified infested samples (false negative)”.
I think this should be clarified because it is confusing.
Author Response
General Comments
The manuscript is well structured, clear and the conclusions are in accordance with the objectives. The analysis of the data is correct as well as the discussion of the results.
There is one issue that is not clear:
Recall has been defined in line 272 as " Recall is the true positive rate that relates to the number of pixels belonging to infested area that were predicted positive and those that the model incorrectly does not capture as infested. ".
Response: The definition of recall or sensitivity was modified by adding “true positive” and “false negative” terms (lines 304-311).
But in line 381, it states: "The most important metric for the detection of a pest of concern such as CM is recall or sensitivity which reflects the amount of incorrectly classified infested samples (false negative)”.
I think this should be clarified because it is confusing.
Response: Clarified the recall definition (lines 309-311).

Reviewer 3 Report
The manuscript is interesting and focusing on insect presence detection on apple. The text is easy to follow, but English must be improved in terms of accuracy. Many phrases reflect how people speak, but scientific paper should be more strict.
Please find detailed comments in the attached PDF.
Real time and fast HSI optimization shall be discussed more in introduction, such as DOI: 10.1556/progress.3.2007.4
Comments:
- many times authors use "a" or "an", but these have the meaning of one or one of. For example, "a maximum" does not exist, "the maximum" or "local maximum" might be.
- L75-77: machine vision can work gray scale too, not only color (RGB). Please correct.
- L83: HSI can find the location of infection is very general statement. Please specify why HSI can do it better than others? Maybe not this is the best advantage, or authors may highlight spatial information.
- L88: HSI can be used to measure size and color of objects, but why shall do? Cheaper solutions can do that. Please reconsider and add reference in case want to keep.
- L95-98: please add reference to application list.
- L100-103 and 109-110 report different classification accuracy for the same study. Please use the best score or provide details about the difference.
- L126: please correct structure, reducing prediction is probably not aim in any study.
- L129-131: incorrect, please correct. LDA and QDA cannot be used for spectra analysis. In case variables exceed the number of samples, DA get unbalanced and loose power. This is the reason why PLS-DA is used instead.
- L150-151: please provide more information about organic cultivars. How those cultivars are recognized organic?
- L154: how apples were disinfected in the solution? Immersed?
- L157: "instar" please use words common in English or available in Thesaurus, such as linked Merriam-Webster.
- L177: please use degree symbol instead of letter "o".
- L179-180: file name extension and file format are two different things. Please clarify.
- L223: Savitzky-Golay can adjust boxcar window and coefficients differ according to the runlength. Please specify.
- L239-243: obvious information, please remove.
- L271: F1-score is not defined. Readers not familiar with this method cannot understand. Please explain or provide definition.
- L284: normalization method was not mentioned. According to presented figure, it was not SNV. Please provide more details about the algorithm. If only offset was corrected, cannot call normalization.
- Figure 4: this cloud is hard to understand, please use projected figures instead (with class ellipses).
- Table 1,2: please remove citation from caption as these are results from present study. Indicate inside the table results from another source.
- Please use "validation" instead of testing. It is more common and easy to understand.
- L380,385: please use the same method name.
- L395: "relatively reached" is impossible. Please use another term.
- Table 5: reported optimal selection cannot be result of SFS. As SFS was told to add new wavelengths step-by-step (L390-392), bigger optimal set must include the smaller. Please clarify.
- L418-423: the optimal set of 22 wavelengths is smaller than the whole spectra, but no data was presented to compare speed or any computational cost.
- Conclusions shall highlight that reported success rates belong to validation or calibration/training.

Author Response
General Comments
The manuscript is interesting and focusing on insect presence detection on apple. The text is easy to follow, but English must be improved in terms of accuracy. Many phrases reflect how people speak, but scientific paper should be more strict.
Response: The entire manuscript was thoroughly revised to address the grammatical and non-technical writing language issues the reviewer raised.
Please find detailed comments in the attached PDF.
Real time and fast HSI optimization shall be discussed more in introduction, such as DOI: 10.1556/progress.3.2007.4
Response: Discussed (lines133-147).
Comments:
- many times authors use "a" or "an", but these have the meaning of one or one of. For example, "a maximum" does not exist, "the maximum" or "local maximum" might be.
Response: We checked the whole manuscript and made correction where we thought it makes grammatical sense to do replace “a”/”an” with “the”.
- L75-77: machine vision can work gray scale too, not only color (RGB). Please correct.
Response: Correction made (line 80).
- L83: HSI can find the location of infection is very general statement. Please specify why HSI can do it better than others? Maybe not this is the best advantage, or authors may highlight spatial information.
Response: More explanation has been provided to justify HSI approach (lines 85-90).
- L88: HSI can be used to measure size and color of objects, but why shall do? Cheaper solutions can do that. Please reconsider and add reference in case want to keep.
Response: modified and corrected (line 94).
- L95-98: please add reference to application list.
Response: Reference added (105).
- L100-103 and 109-110 report different classification accuracy for the same study. Please use the best score or provide details about the difference.
Response: The classification accuracies for the same study are different because the results were from the two different methods used in the same study. This was explained (lines 115-117).
- L126: please correct structure, reducing prediction is probably not aim in any study.
Response: Thank you for catching this typo, the structure of the sentence was corrected (lines 137-138).
- L129-131: incorrect, please correct. LDA and QDA cannot be used for spectra analysis. In case variables exceed the number of samples, DA get unbalanced and loose power. This is the reason why PLS-DA is used instead.
Response: Corrected. Also new references added for the preference of PLS-DA and ensemble methods for spectral data (lines 149-155).
- L150-151: please provide more information about organic cultivars. How those cultivars are recognized organic?
Response: The samples were USDA certified organic apples purchased from local market. Detail provided. (line 165)
- L154: how apples were disinfected in the solution? Immersed?
Response: They were washed in the disinfection solution and then rinsed with water (line 170).
- L157: "instar" please use words common in English or available in Thesaurus, such as linked Merriam-Webster.
Response: Replaced (lines 172).
- L177: please use degree symbol instead of letter "o".
Response: Corrected (line 176).
- L179-180: file name extension and file format are two different things. Please clarify.
Response: Clarified (lines 195-196).
- L223: Savitzky-Golay can adjust boxcar window and coefficients differ according to the runlength. Please specify.
Response: Clarified (lines 238-239)
- L239-243: obvious information, please remove.
Response: While we reasoned with the reviewer here. A research article serves many purposes, including reinforcing principles that are known. Readers of scientific articles are at different stages in their career, a reminder of some baseline principles have a way of reinforcing their application and understanding by the reader.
- L271: F1-score is not defined. Readers not familiar with this method cannot understand. Please explain or provide definition.
Response: Definition of F1 provided (lines: 301-302).
- L284: normalization method was not mentioned. According to presented figure, it was not SNV. Please provide more details about the algorithm. If only offset was corrected, cannot call normalization.
Response: We refer the reviewer to lines 238 and 242. We have provided more details about the algorithm.
- Figure 4: this cloud is hard to understand, please use projected figures instead (with class ellipses).
Response: Modified! The new plot is for the first two PCs computed from the dataset built from the mean spectral of each apple instead of each pixel (lines 360).
- Table 1,2: please remove citation from caption as these are results from present study. Indicate inside the table results from another source.
Response: Removed!
- Please use "validation" instead of testing. It is more common and easy to understand.
Response: Replaced.
- L380,385: please use the same method name.
Response: Corrected
- L395: "relatively reached" is impossible. Please use another term.
Response: Corrected (lines 441).
- Table 5: reported optimal selection cannot be result of SFS. As SFS was told to add new wavelengths step-by-step (L390-392), bigger optimal set must include the smaller. Please clarify.
Response: The SFS was used for optimal selection in each subset of feature numbers. But the author came into conclusion about the reason why the wavelengths are different in different iteration of subsets such that it is due to randomness associated with cross-validation used for the data split at each run. Since the objective of the feature selection in this study was to show that even the smaller data set can give good results as the initial big dataset and also the selected wavelengths are in the same range (for example the main ranges of 970-985, 1040-1080, 1140-1350 and 1620-1670 nm which are selected in the 5 features run are recognizable in the bigger sets for Fuji cultivar), the wavelength selection section was included in this form.
- L418-423: the optimal set of 22 wavelengths is smaller than the whole spectra, but no data was presented to compare speed or any computational cost.
Response: That is right. The smaller datasets with only 22 wavelengths will most probably reduce computational costs due to reduced data complexity and number of features, but no cost evaluation was intended and carried out in this current study.
- Conclusions shall highlight that reported success rates belong to validation or calibration/training.
Response: The accuracies reported in the conclusion section belong to the validation or test not training. It was further highlighted.

Round 2
Reviewer 1 Report
The paper was corrected as suggested and now it could be accepted.